# ENSO phase transition enables prediction of winter North Atlantic Oscillation one year ahead

Kiwook Kim [1], Myong-In Lee [1] ✉, Adam A. Scaife [2,3] & Doug M. Smith [2]

The winter North Atlantic Oscillation (NAO) is a dominant mode of climate variability affecting temperature and precipitation across the Northern Hemisphere, yet its prediction at seasonal-to-decadal (S2D) lead times remains challenging. Here, using multi-year hindcasts from a multi-model ensemble initialized on 1 November for 1962–2019, we show that NAO skill one year ahead improves significantly when the El Niño–Southern Oscillation (ENSO) undergoes a phase transition next year. This improvement is linked to the northward propagation of anomalous atmospheric angular momentum, which dynamically organizes the NAO and is captured in reanalysis and models. During ENSO transition years, prediction skill increases with ensemble size, and when more than 10 members are used, the forecasts display the signal-to-noise paradox. These findings highlight the potential for enhanced one-year NAO predictability when ENSO transitions are present and large ensemble sizes are used in S2D prediction systems, given the skillful prediction of ENSO phase transitions at one-year lead times by multi-model ensembles.

The winter North Atlantic Oscillation (NAO) is the dominant mode of atmospheric variability that drives various weather and climate anomalies in the Northern Hemisphere[1–7]. Interannual variability of NAO strongly influences regional extremes, particularly across Europe[8–10], underscoring the need for skillful forecasts to support socio-economic risk management[11,12]. In response, subseasonal-to-seasonal (S2S) and seasonal forecasts (typically up to six months lead time) have been widely explored. Most existing studies have focused on 1–3-month lead predictions of the winter NAO (or the Arctic Oscillation, AO), identifying multiple potential sources of predictability[13–15]. Remote precursors such as the El Niño–Southern Oscillation (ENSO), sea ice, and the stratospheric polar vortex have been particularly emphasized for their ability to provide predictability of Atlantic dynamics in models (e.g.[16–19]).

However, extending NAO prediction to longer lead times—seasonal-to-interannual (S2I) or seasonal-to-decadal (S2D) scales—remains challenging[20–24]. Nevertheless, long-range forecasts are increasingly recognized as essential for proactive planning in sectors such as agriculture, energy, and infrastructure[25–27]. While decadal prediction systems have shown significant skill in simulating multi-year mean NAO variability[22,23], predicting the interannual winter NAO index remains elusive. A few systems have demonstrated significant skill in predicting the NAO one year ahead[22,28] and they have shown that importance of the ensemble size (40 members) and incorporating external dynamical factors such as sea surface temperature (SST) oceanic variability, ENSO, North Atlantic Tripolar SST (NATS), sea ice, and a strong polar vortex (SPV) through remote teleconnections can enhance NAO prediction skill. Yet, both the mechanism and its generalization across models remain to be demonstrated. To address this limitation, we adopt a Multi-Model Ensemble (MME) approach, using long-term prediction systems such as the Decadal Climate Prediction Project (DCPP)[29] and the Community Earth System Model 2-Seasonal to Multiyear Large Ensemble (CESM2-SMYLE)[30], to investigate the predictability of the NAO during the second winter after initialization (i.e., one-year lead; hereafter LY1).

Recent evidence suggests that NAO variability one year ahead may not only reflect concurrent external forcings but also delayed dynamical linkages[31]. In particular, strong ENSO variability, especially when

[1]Department of Civil, Urban, Earth and Environmental Engineering, Ulsan National Institute of Science and Technology, Ulsan, South Korea. [2]Met Office Hadley Centre, Exeter, United Kingdom. [3]Department of Mathematics and Statistics, University of Exeter, Exeter, United Kingdom. ✉e-mail: milee@unist.ac.kr

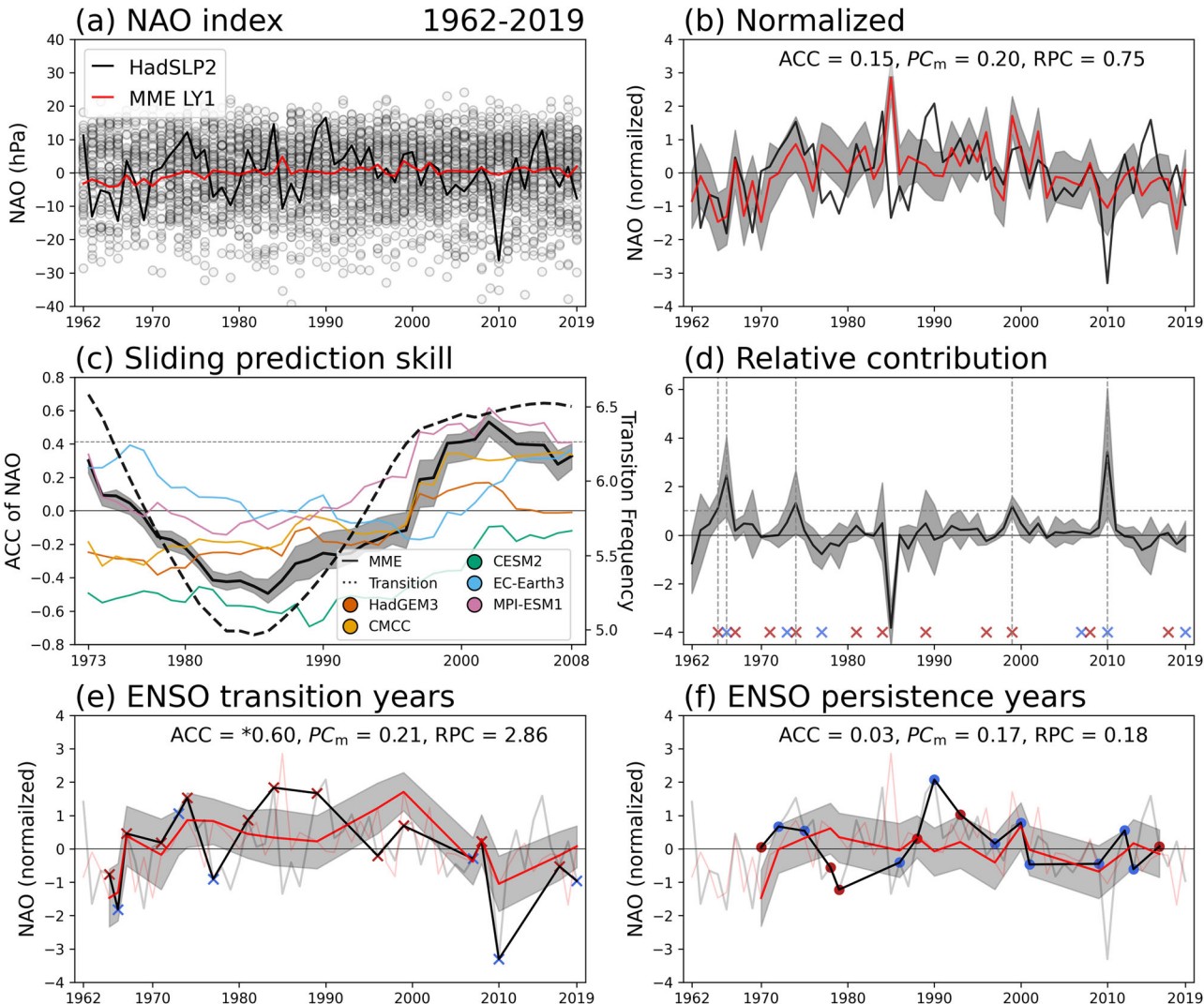

**Fig. 1 | Periods of increased interannual predictability of the North Atlantic Oscillation coincide with frequent El Niño–Southern Oscillation transitions. a** January–February (JF) mean North Atlantic Oscillation (NAO) index from HadSLP2 (black) and the multi-model ensemble (MME) at lead year 1 (LY1; red) for 1962–2019. Grey dots indicate individual ensemble members (total = 70). **b** Normalized NAO indices corresponding to (a), with ensemble spread shown by shading (10th – 90th percentiles). The anomaly correlation coefficient (ACC), predictable component in models ($PC_m$), and ratio of predictable components (RPC) are indicated. **c** The 23-year sliding ACC between HadSLP2 and individual prediction systems (colored lines) and the MME (black). The dashed black line shows the 23-year running mean observational El Niño–Southern Oscillation

(ENSO) transition frequency, defined from a binary time series (1 for transition years, 0 otherwise). The horizontal dashed line denotes the 5% significance level. **d** Relative contribution (RC) of individual years to the MME NAO prediction skill. The horizontal dashed line marks RC = 1. Red and blue crosses indicate El Niño and La Niña transition years, respectively. **e, f** Composite normalized NAO indices for ENSO transition years (**e**; $n = 17$) and ENSO persistence years (**f**; $n = 16$). Shading denotes ensemble spread (10th – 90th percentiles). Asterisk indicates statistical significance at the 5% level. See Methods for definitions of ENSO transition and persistence. In (b–f), the 10th–90th percentile range was derived from 1000 bootstrap resamples.

accompanied by a phase transition between El Niño and La Niña, appears to influence NAO amplitude with a one-year lag. This delayed effect can be attributed to tropical ENSO forcing and the slow pole-ward propagation of atmospheric angular momentum (AAM) anomalies[32–34].

Building on these insights, this study re-evaluates the role of ENSO as a delayed external precursor in long-range prediction systems using an MME framework. We formulate the hypothesis that ENSO transitions enhance the predictability of the winter NAO one year in advance. We demonstrate both the statistical and dynamical linkages between ENSO variability and the NAO with a one-year lag, empha-sizing the distinct role of ENSO phase transitions. Discussion con-cludes with implications for improving dynamical ensemble prediction systems.

## Results

### Skilful NAO prediction one year ahead associated with ENSO transition

Figure 1 shows the LY1 prediction skill of the NAO index using 70 ensemble members from the MME during 1962–2019. The ensemble spread of NAO anomalies is large, and the MME severely under-estimates the observed amplitude of the index (Fig. 1a). The anomaly correlation coefficient (ACC) skill over the full period is weak (ACC = 0.15, $p = 0.27$), and the relative predictable component (RPC) remains below 1, because the predictable component in models ($PC_m = 0.20$) is larger than that inferred from observations ($PC_o$ or ACC), indicating limited predictability (Fig. 1b; see Methods). This contrasts with DePreSys3, which demonstrated higher skill (r = 0.42, $p = 0.01$ for 1982–2016)[28].

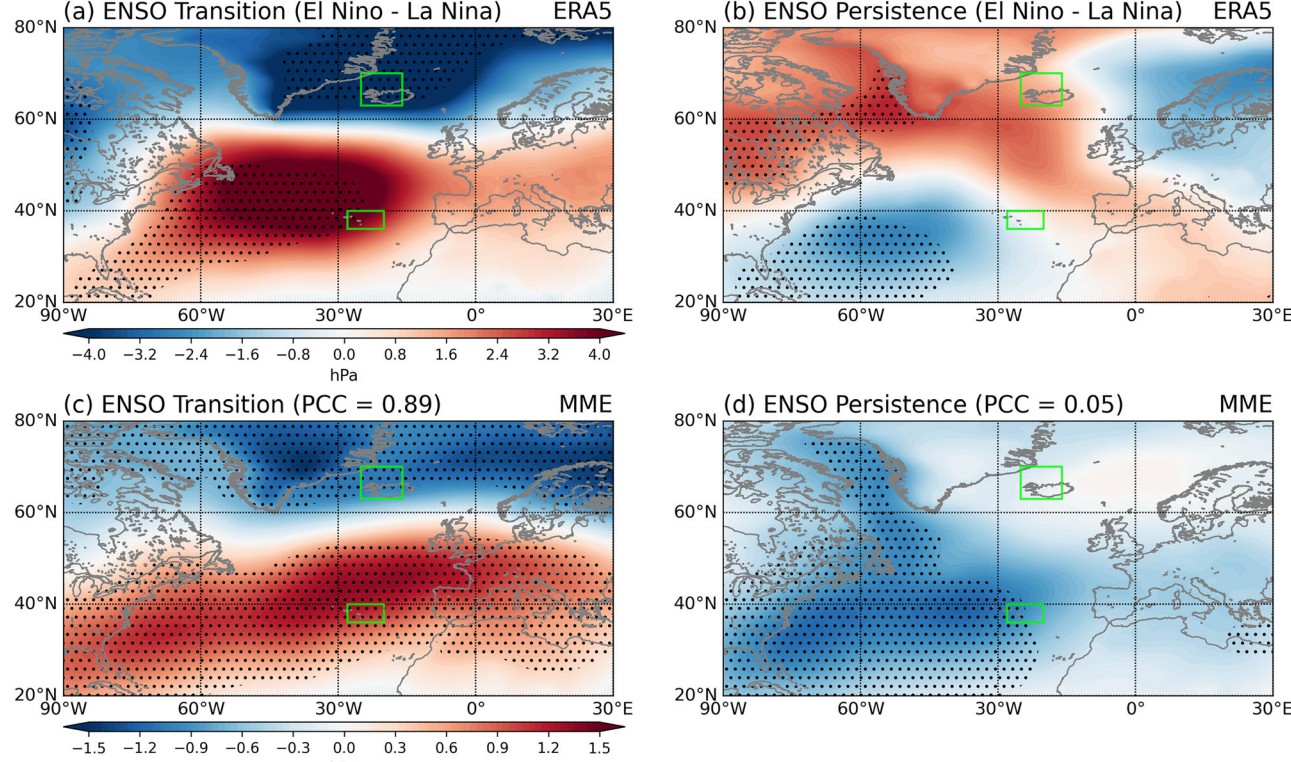

**Fig. 2 | Enhanced interannual predictability of the Atlantic sector during El Niño–Southern Oscillation transition years.** Composite differences in January–February (JF) mean sea level pressure (SLP) anomaly (shaded, hPa) between El Niño and La Niña conditions for **a** El Niño–Southern Oscillation (ENSO) transition years and **b** persistence years in ERA5. **c**, **d** Corresponding composite differences from the multi-model ensemble (MME). Pattern correlation coefficients (PCC) are calculated over 20°–80°N, 60°W–30°E. Green boxes denote the North Atlantic Oscillation (NAO) definition regions (Azores and Iceland). Stippling denotes statistical significance at the 95% confidence level, based on a two-sample t-test for observations and 1000 bootstrap resamples for the MME (see Methods).

A 23-year sliding window of ACC reveals that NAO prediction skill increased markedly in the late 1990s for both the MPI model and the MME, with other models also showing upward trends (Fig. 1c). Following the ENSO-NAO delayed teleconnection framework[34], we examine whether this improvement is linked to the frequency of ENSO phase transitions (see Methods). The 23-year sliding-mean ENSO transition frequency (dashed line in Fig. 1c) exhibits peaks in the early 1970s and after the mid-1990s, closely aligned with the 23-year running-mean NAO skill ($r = 0.94$, $p < 0.01$). This suggests that ENSO transitions after initialization are a major driver of improved LY1 NAO prediction. The relative contribution (RC; see Methods) analysis further supports this relationship: all years with RC over 1 coincide with ENSO transition years (vertical gray dashed lines in Fig. 1d; 1965, 1966, 1976, 1999, and 2010). Three of these transitions were from El Niño to La Niña, and two were the reverse.

Separating ENSO transition and persistence years reveals a clear contrast in NAO prediction skill (ACC in Fig. 1e, f). During transition years, observed NAO variability is significantly larger ($\sigma_{NAO} = 10.53$ hPa) than during persistence years ($\sigma_{NAO} = 6.34$ hPa; F-test, $p < 0.05$). Moreover, NAO prediction skill is remarkably higher during transition years, with ACC = 0.60 ($p = 0.012$) and RPC = 2.86 ($p < 0.001$), indicating the presence of a signal-to-noise paradox (SNP)[35]. In this case, real-world predictability exceeds model-estimated skill (RPC > 1). Although only the MPI model and the MME individually achieve statistically significant skill (Fig. S1a), the consistent enhancement across the models during ENSO transitions demonstrates that tropical phase changes robustly reinforce NAO predictability one year ahead. In contrast, persistence years show little to no skill (Fig. 1f). Using seasonal DJF-mean NAO indices yields qualitatively identical results, although with lower prediction skill than the JF-mean definition

(Fig. S2), confirming that our conclusions are not sensitive to the specific winter season definition (see Methods).

This dependence on ENSO phase transitions highlights a source of long-lead NAO predictability beyond the initialized season. The emergence of the SNP over the North Atlantic, previously documented in 1–3month forecasts[15,23,28,35,36], is now evident at a one-year lead (consistent with Dunstone et al., 2016[28]). This underscores the dynamical connection between tropical ENSO transitions and extratropical NAO variability operating on S2D time scales[34].

To further identify which node of the NAO contributes to enhanced prediction skill during ENSO transition years, we analyze the spatial structure of composite sea level pressure (SLP) anomalies one year after El Niño and La Niña events (Fig. 2). In ERA5, El Niño-La Niña transition years exhibit a clear positive-phase NAO pattern (Fig. 2a), whereas persistence years show little to no NAO signal (Fig. 2b). This indicates that ENSO transitions generate NAO variability through remote teleconnections. This is consistent with the constructive interference from lagged and simultaneous signals shown in Scaife et al. (2024)[34] and suggests that there are structured NAO signals to predict during ENSO transition years.

This is consistent with composites of the observed NAO change anomaly one year later (ΔNAO), which show that when an ENSO event occurs in the preceding year and subsequently transitions, both El Niño and La Niña cases are associated with significant NAO anomalies. By contrast, no statistically significant NAO changes are observed when the preceding year is a neutral state (|ENSO| < 0.5 K) or when ENSO persists in the following year (Table S1).

The MME reproduces a similar dipole structure during transition years (Fig. 2c), with strong agreement with observations (PCC = 0.89). Significant SLP anomalies appear over the Azores in both El Niño and

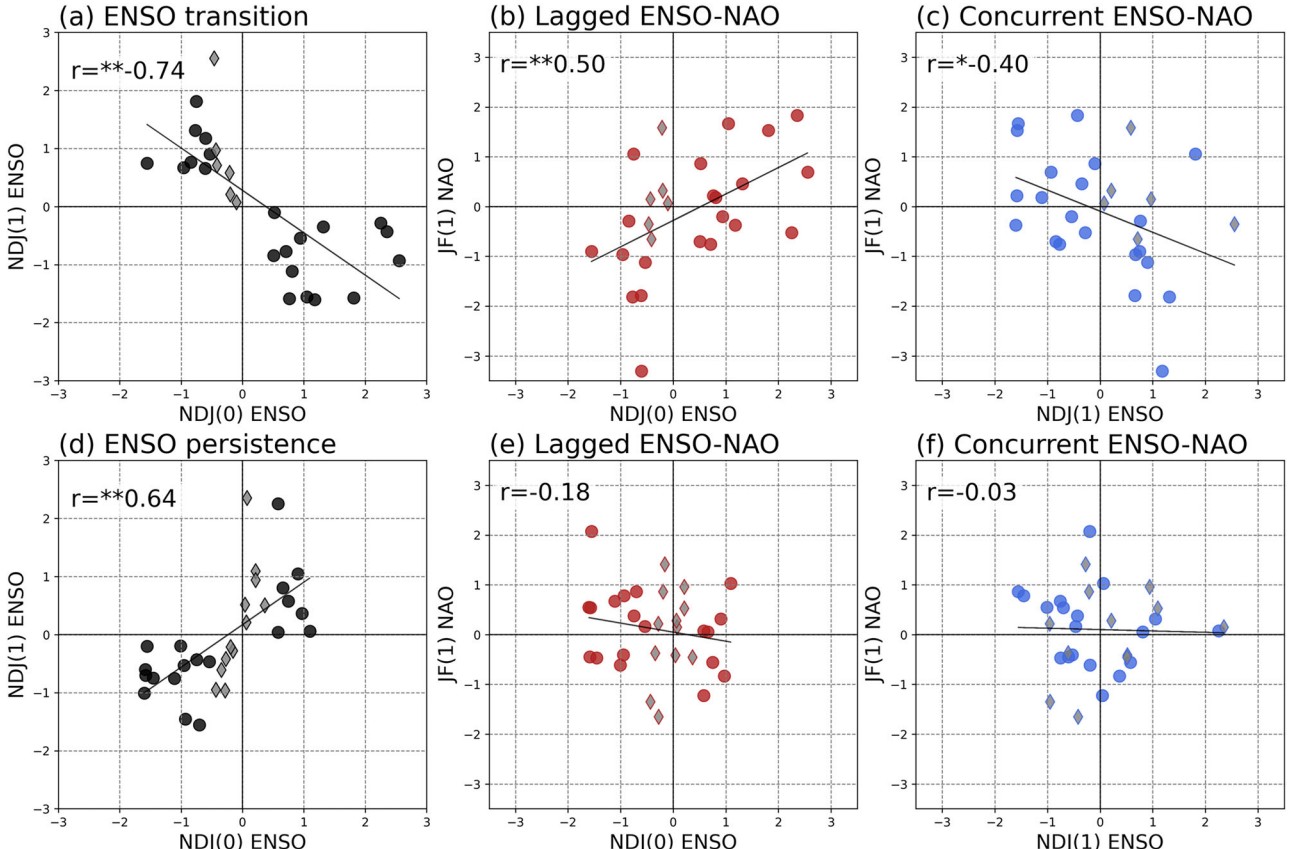

**Fig. 3 | Transition years produce stronger North Atlantic Oscillation anomalies.**
**a** Observed relationship between the November–December–January of year 0 (NDJ(0)) mean El Niño–Southern Oscillation (ENSO) index and the November–December–January of year 1 (NDJ(1)) mean ENSO index during ENSO transition years. **b** One-year lagged relationship between the NDJ(0)-mean ENSO index and the January–February of year 1 (JF(1)) mean North Atlantic Oscillation (NAO) index during transition years. **c** Concurrent relationship between the NDJ(1)-mean ENSO index and the JF(1)-mean NAO index during transition years. **d**–**f** Same as (**a**–**c**), but for ENSO persistence years. Circles indicate strong ENSO years, while diamonds denote weak ENSO transition or persistence years in the first winter (see Methods). Correlation coefficients (r) are shown in each panel, and ** and * denote statistical significance at the 1% and 5% levels, respectively.

La Niña transition years (Fig. S3). In contrast, persistence years show no dipole structure; instead, the MME produces a monopole-like negative anomaly centered over the North Atlantic with no resemblance to the observed anomaly (PCC = 0.05, Fig. 2d). The interannual predictions clearly show higher skill during ENSO transition years.

These results demonstrate that the delayed ENSO–NAO teleconnection mechanism during transition years is robust in both observations and models. In contrast, persistence years fail to produce comparable NAO responses one year ahead and are poorly predicted, consistent with weaker predictable signals and destructive interference between lagged and concurrent teleconnections during ENSO persistence years[34]. Our findings are consistent with ENSO transitions being a major driver of year ahead prediction skill for the NAO.

**Lagged dynamics of ENSO on NAO and the role of AAM**
To examine whether the ENSO-NAO relationship depends on ENSO evolution, we analyzed scatter distributions of observed indices (Fig. 3). ENSO events are categorized into transition and persistence types based on the following year's ENSO phase (Fig. 3a, d; see Methods), which reveals distinct linear relationships with subsequent winter ENSO. However, there is a notable difference in ENSO variability. El Niño persistence events are weaker with most indices not exceeding 1 in the first winter, while La Niña events show stronger anomalies (black dots in Fig. 3d). This asymmetry reflects well-known ENSO characteristics under persistence regimes, including central-Pacific (CP) El Niño and multi-year La Niña patterns[37,38].

During ENSO transition years, ENSO exhibits distinct linear relationships with the following winter NAO, whereas ENSO persistence years result in only weak NAO anomalies (red, blue dots in y-axis). The one-year lagged ENSO-NAO correlation is evident during transition years but absent during persistence years (red dots in Fig. 3b, e). Specifically, transition years show a positive lagged correlation, while the concurrent correlation is negative (blue dots). This pattern is consistent with Scaife et al. (2024)[34], which suggested that positive (negative) atmospheric angular momentum (AAM) anomalies linked to prior-year El Niño (La Niña) generate lagged positive (negative) NAO signals, whereas concurrent Rossby wave forcing from La Niña (El Niño) induces positive (negative) NAO-like anomalies. These dual mechanisms may act synergistically, amplifying NAO variability during ENSO transitions (Fig. 1).

Spatial patterns of SST anomalies provide further support (Fig. S4). In transition years, a strong eastern-Pacific (EP)-like El Niño develops one year prior, while a CP-like La Niña emerges in the concurrent winter. These phase shifts are consistent with the Quasi-Quadrennial (QQ) mode of ENSO (3–7-year periodicity)[39], suggesting that the transition in QQ mode is also a key driver of lagged ENSO–NAO teleconnection dynamics.

The MME simulations broadly reproduce the observed distinction between transition and persistence regimes (Fig. S5). However, models underestimate the lagged ENSO–NAO correlation during transition years (r = 0.36, p < 0.05), and overestimate the concurrent relationship during both types of years (r = −0.47 in transition, −0.54 in persistence

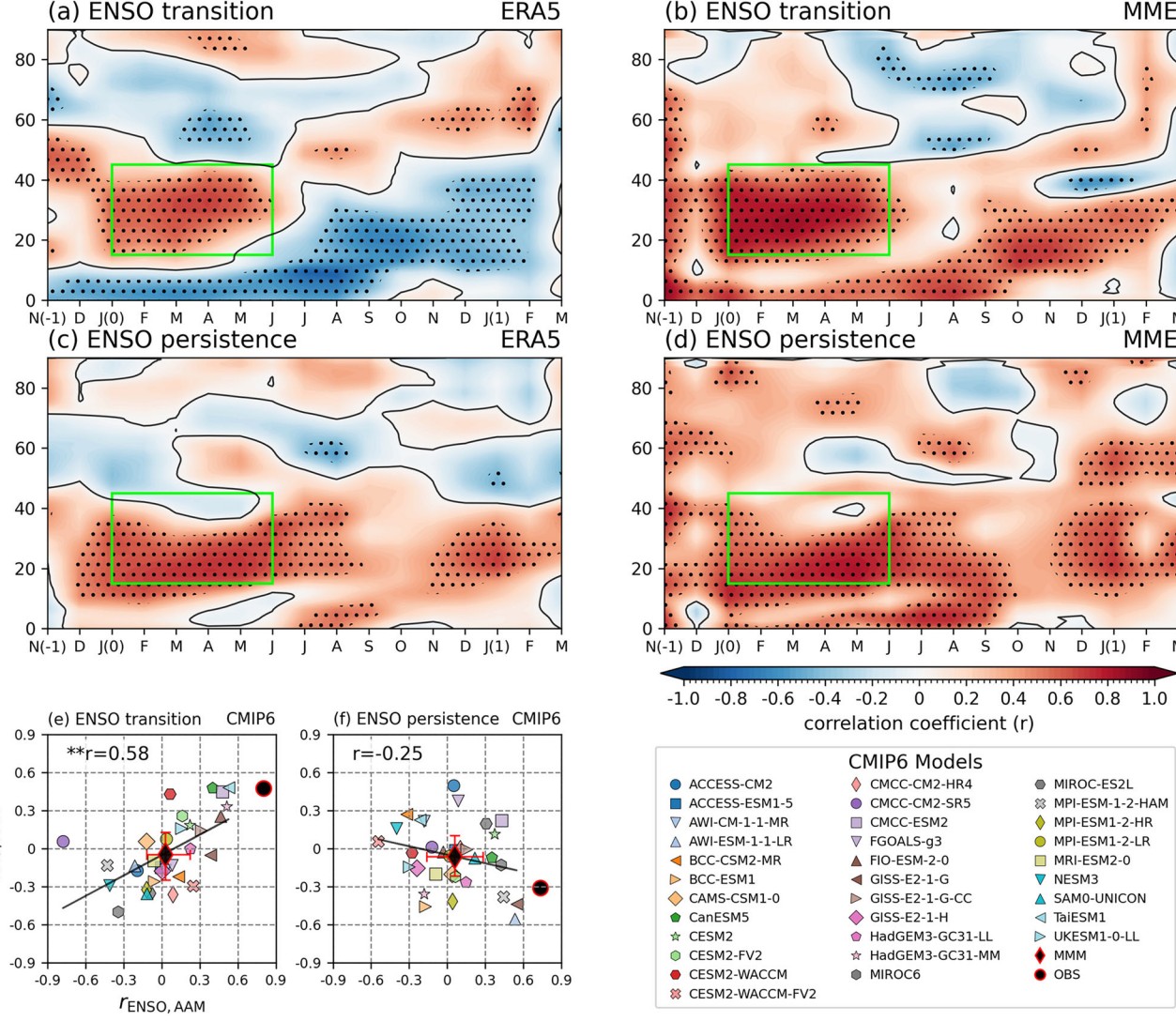

**Fig. 4 | Poleward-propagating atmospheric angular momentum anomalies emerge only in El Niño–Southern Oscillation transition years. a** Monthly lagged correlation between the November–December–January of year 0 (NDJ(0)) mean El Niño–Southern Oscillation (ENSO) index and zonal-mean atmospheric angular momentum (AAM) anomalies ($10^{24}$ kg m$^2$ s$^{-1}$ per latitude) in the Northern Hemisphere during ENSO transition years, based on ERA5. **b** The prediction skill of AAM in the multi-model ensemble (MME). **c, d** Same as in (**a, b**) but for ENSO persistence years. Stippling denotes statistical significance at the 90% confidence level. Significance for the MME is estimated using 1000 bootstrap resamples (see Methods).

**e, f** Scatter plots across CMIP6 historical simulations (1962–2014) showing the relationship between the correlation of NDJ(0)-mean ENSO and AAM (x-axis, averaged over the green box; 15–45°N, January–June (JFMAMJ)) and the correlation between AAM and the January–February of year 1 (JF(1)) mean North Atlantic Oscillation (NAO) index (y-axis), for ENSO transition and ENSO persistence years, respectively. The black circle denotes observations, and the black diamond indicates MME. Error bars represent the interquartile range (25th–75th percentiles) across models. Correlation coefficients (r) are indicated in each panel, and ** denotes significance at the 1% level.

years, $p < 0.01$). This overestimation likely reflects a systematic model bias toward simulating concurrent Rossby wave teleconnections from ENSO forcing at the expense of the lagged connections. Such biases are well documented in seasonal prediction systems[28].

To further understand how ENSO affects the NAO one year later, we examined the role of AAM as a lagged dynamical bridge. Previous work[31,34] highlighted the slow poleward propagation of AAM anomalies induced by tropical ENSO forcing as a key mechanism. Observations confirm this process. During ENSO transition years, El Niño events are associated with strong positive AAM anomalies in boreal first winter (LY0), which gradually propagate poleward (Fig. 4a). By the following second winter (LY1), a dipole pattern emerges, with negative anomalies in the tropics and positive anomalies at higher latitudes, consistent with eddy feedback and wave activity flux favoring a positive NAO[31]. In persistent years, however, AAM anomalies are weak, confined to the low latitudes, and fail to propagate poleward (Fig. 4c). This indicates

that a robust poleward AAM signal is evident only during ENSO transition years.

The MME prediction skill of AAM supports long-range dynamical signals. For transition years, the MME shows significant AAM predictability up to six months after initialization, particularly through the following spring and into LY1 winter (Fig. 4b). In contrast, persistence years show no meaningful AAM prediction skill in the initial winter, with only weak and inconsistent skill emerging in spring (Fig. 4d). Northward propagating anomalies in the second winter (LY1) are well captured for transition years.

CMIP6 historical simulations further confirm these findings (Fig. 4e, f). In transition years, the NDJ(0)-mean ENSO index correlates significantly with the first winter-spring averaged AAM index (green box in Fig. 4a; 15°–45°N, JFMAMJ), which in turn correlates with the subsequent NAO ($r = 0.58$, $p < 0.01$). This confirms that early AAM anomalies triggered by ENSO serve as a key source of lagged NAO

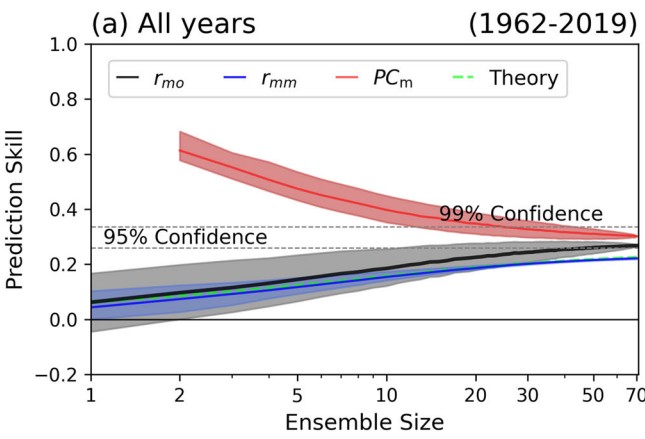

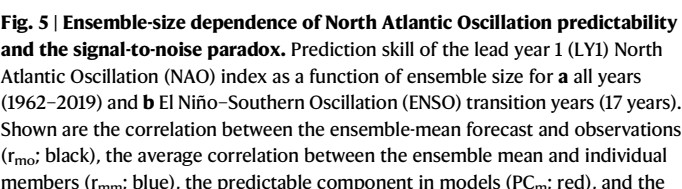

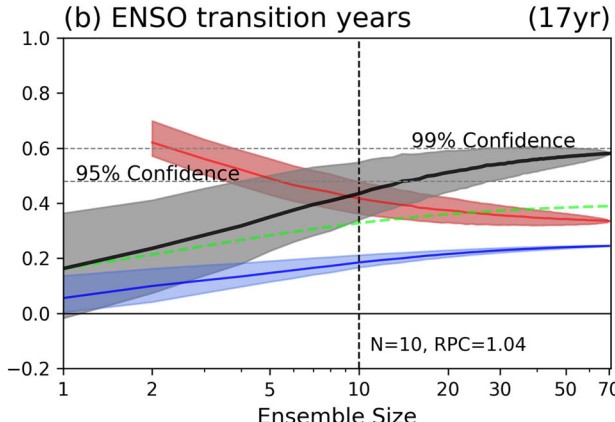

**Fig. 5 | Ensemble-size dependence of North Atlantic Oscillation predictability and the signal-to-noise paradox.** Prediction skill of the lead year 1 (LY1) North Atlantic Oscillation (NAO) index as a function of ensemble size for **a** all years (1962–2019) and **b** El Niño–Southern Oscillation (ENSO) transition years (17 years). Shown are the correlation between the ensemble-mean forecast and observations ($r_{mo}$; black), the average correlation between the ensemble mean and individual members ($r_{mm}$; blue), the predictable component in models ($PC_m$; red), and the theoretical estimate of the anomaly correlation coefficient (ACC; green dashed). Shading denotes the interquartile range (25th–75th percentiles) estimated from 10,000 bootstrap resamples without replacement (see Methods). Horizontal dashed lines indicate the 95% and 99% confidence levels. In (**b**), the vertical dashed line marks the minimum ensemble size (N = 10) at which $r_{mo}$ exceeds $PC_m$, indicating the emergence of the signal-to-noise paradox (RPC = 1.04).

variability. Notably, CMIP6 models systematically underestimate both the ENSO–AAM and AAM–NAO correlations compared to observations, and only 11 models capture positive correlations along both axes (Fig. 4e). In transition years, while observations show significant positive linkages (black circle), several models even simulate opposite-signed relationships, highlighting the limited ability of current climate models and multi-model mean (MMM; black diamond) to capture the delayed ENSO teleconnection. In persistent years, the model spread is large, the correlation is insignificant (r = −0.23, p > 0.05), and no robust ENSO–AAM–NAO linkage is found.

Overall, these results demonstrate that the poleward propagation of AAM induced by ENSO transitions is the primary dynamical mechanism driving delayed NAO variability. Accurate representation of this process in prediction systems is now a priority for model development and is essential for enhancing NAO predictability at a one-year lead.

### Ensemble dependence and overcoming the signal-to-noise in NAO predictions

As shown in the previous section, the poleward propagation of AAM provides a dynamically structured pathway linking ENSO to NAO variability. An important question is how this mechanism is reflected in prediction systems, and how prediction skill depends on ensemble size. To address this, we evaluated the sensitivity of NAO skill to the number of ensemble members by comparing the real-world correlation ($r_{mo}$), the model-world correlation ($r_{mm}$), and the $PC_m$ (Fig. 5). Over the full analysis period (1962–2019), one-year NAO prediction skill remains weak. The $r_{mo}$ (= $PC_o$) never exceeds the model-based $PC_m$, indicating limited skill of the models in simulating observed NAO variability overall (Fig. 5a).

In contrast, during ENSO transition years, prediction skill improves markedly as ensemble size increases (Fig. 5b). Once the number of ensemble members exceeds ~10, the $r_{mo}$ surpasses the $PC_m$, revealing the emergence of the SNP. The growth rate of $r_{mo}$ outpaces that of $r_{mm}$, suggesting that the real world contains more predictable signal than is represented by the models. This discrepancy is particularly pronounced in transition years.

Comparison with theoretical estimates further emphasizes this point. While the theoretical ACC (green dashed line; see Methods) does not reach statistical significance even with 70 ensemble members, the observed $r_{mo}$ exceeds the 95% confidence level with only

~15 members and approaches the 99% level with more than 70 members. This result suggests that internal atmospheric variability is more systematically organized during ENSO transitions in the real world than in the model world.

Overall, these findings demonstrate that under ENSO transition conditions, increasing ensemble size yields substantial improvement in one-year NAO prediction skill, exceeding theoretical expectations and confirming the practical benefits of ensemble expansion when favorable dynamical mechanisms are active. It also indicates that the AAM propagation mechanism may be intimately involved in the SNP.

## Discussion

This study shows that ENSO phase transitions substantially enhance the one-year-ahead predictability of the winter NAO, based on MME long-range prediction experiments. The observed lagged ENSO–NAO linkage is reproduced in both spatial and temporal structures of simulated NAO variability. In particular, the poleward propagation of AAM anomalies triggered by ENSO transitions is confirmed as a key mechanism that amplifies winter NAO variability and enhances predictability, consistently represented in initialized prediction systems and CMIP6 models. However, the spread across historical simulations highlights the limited ability of current coupled climate models to reproduce the delayed ENSO teleconnection via the ENSO-AAM-NAO pathway, which may partly account for the muted NAO predictability across multi-model ensembles.

During ENSO transition years, the observed ACC exceeds the model-estimated $PC_m$ (RPC > 1), indicating the presence of a signal-to-noise paradox (SNP). This paradox strengthens as ensemble size increases, with real-world predictability surpassing theoretical ensemble-based expectations. These results indicate that ENSO transitions structurally organize internal atmospheric variability, thereby enhancing predictability at interannual lead times.

Although other drivers, such as stratospheric processes and midlatitude feedback, may also contribute to NAO variability, this study emphasizes the delayed tropical forcing pathway. The timescale and spatial progression of AAM anomalies support a mechanistic link between ENSO transitions and NAO predictability one year ahead, extending beyond concurrent or seasonal-scale influences.

The MME results indicate that NAO prediction skill is enhanced during ENSO phase transitions. For this result to be practically meaningful, it is essential to assess how well climate models can predict ENSO

phase evolution (see Methods). The MME exhibits statistically significant long-range ENSO prediction skill through the LY1 winter despite the spring predictability barrier (Fig. S6). We further examine one-year-ahead tercile prediction skill conditioned on the initial ENSO state (Fig. S7). We demonstrated that these categories are themselves highly accurately predicted, confirming that the conditioning reflects realistic forecast skill rather than an artificial perfect-model assumption.

Overall, these findings highlight the critical role of combining ENSO phase transition signals with large-ensemble strategies to improve NAO forecasts on S2I and S2D timescales. The results also underscore the importance of accurately representing internal variability and external forcings in future interannual-to-decadal prediction systems.

## Methods

### Model and reanalysis data

We analyzed long-range predictions from four DCPP models (Had-GEM3-GC3.1-DePreSys4, CMCC-CM2-SR5, EC-Earth3, and MPI-ESM1-2-HR), each providing forecasts up to 10 years, along with the CESM2-SMYLE hindcasts covering up to 2 years. All models were initialized on 1 November using full-field assimilation[29] (Table S2). The analysis period covers 1962–2019 for the DCPP models and 1972–2019 for CESM2. A total of 70 ensemble members across all models were used to assess the MME predictability. We defined the first boreal winter after initialization as lead year 0 (LY0) and the second boreal winter as LY1. For example, for a forecast initialized in November 1980, LY0 corresponds to the JF(0) winter of 1981 (2-3 month lead), and LY1 corresponds to the JF(1) winter of 1982 (14-15 month lead). This study focused on evaluating NAO prediction skill during LY1.

For reanalysis, we use sea level pressure (SLP, units: hPa) from the Hadley Centre Sea Level Pressure 2 (HadSLP2)[40] dataset, atmospheric variables from the European Centre for Medium-Range Weather Forecasts Reanalysis version 5 (ERA5)[41], and sea surface temperature (SST) from the Hadley Centre Sea Ice and SST dataset (HadISST1)[42]. All model predictions and reanalysis were regridded to a common 1.25° × 1.25° grid for MME evaluation, while NAO indices were computed on a 5° × 5° grid consistent with HadSLP2.

We further validated our hypothesis using historical simulations from the Coupled Model Intercomparison Project Phase 6 (CMIP6). A total of 32 CMIP6 models were included, all regridded to a common spatial resolution. Model details are provided in Table S3.

### Climate indices definitions

The winter NAO index is defined as the difference in SLP between Iceland (63°–70°N, 25°–16°W) and the Azores (36°–40°N, 28°–20°W), following Jones et al. (1997)[43]. It is calculated using January–February (JF) means, corresponding to the season of maximum variability in the AO and NAO[44–46] (Fig. S8). Previous studies[47,48] have shown that ENSO teleconnections vary within the winter season. In particular, the JF period exhibits a clearer NAO-like annular structure than early winter.

The ENSO index is defined as the first principal component (PC1) of the leading empirical orthogonal function (EOF) of equatorial Pacific SST anomalies (15°S–15°N, 140°E–80°W), averaged over November–January (NDJ). ENSO events are identified when the index exceeds ±0.5 standard deviation (STD). This index exhibits strong agreement with the NDJ-mean Niño3.4 index (r = 0.99), confirming its suitability for characterizing the Quasi-Quadrennial (QQ) periodicity of ENSO[39,49,50] (Fig. S9).

The AAM anomaly is calculated by zonally averaging and vertically integrating the zonal wind field from the lower atmosphere to the model top (up to 1 hPa), consistently across ERA5 and model datasets. AAM is computed as[31]:

$$AAM(\varphi) = \int_{r=a}^{\infty} \int_{\lambda=0}^{2\pi} 2\pi\rho(\Omega r \cos\varphi + \mathbf{U})r^3\cos^2\varphi \, dr \, d\lambda \quad (1)$$

where $\rho$ is atmospheric density, $\lambda$ is longitude, $\varphi$ is latitude, $\mathbf{U}$ is the zonal wind, $r$ is radial distance from Earth's centre, a is the radius of Earth, and $\Omega$ is the mean angular velocity of Earth. All indices (NAO, ENSO, and AAM) are detrended prior to analysis.

### ENSO phase change definitions

ENSO transition years are defined as cases in which the PC1 of ENSO exceeds ±0.5 STD in the first boreal winter and subsequently changes sign in the following NDJ(1) winter. ENSO persistence years are those in which the ENSO phase does not change between the two winters. The years corresponding to each ENSO evolution category are listed in Table S4. To examine the linear relationship and increase the sample size, transition and persistence cases are also included and further classified as weak El Niño or La Niña simply based on the sign of the anomaly in the first winter with diamond symbols in Fig. 3.

To evaluate NAO prediction skill one year ahead (LY1), composites were constructed based on ENSO phase evolution. The MME ENSO index was predicted one year in advance, and only those years in which the ENSO evolution category in LY1 matched between observations and the MME were retained. This conditioning ensures that the composite analysis isolates NAO predictability associated with correctly simulated ENSO phase evolution. Although this approach does not explicitly address the spring predictability barrier[51], the agreement in ENSO category between observations and the MME remains significant prediction skill through LY1, supporting the validity of the composite framework (17 out of 21 transition years and 16 out of 19 persistence years in observations; Fig. S1b, S6).

For the NAO-skill composites, we conditioned on years in which the model-predicted NDJ(1) ENSO category (transition or persistence) agrees with matched observations (Fig. S10). This so-called 'perfect-model' filtering was applied solely as a diagnostic tool to isolate the underlying physical mechanisms. We further demonstrate, using Relative Operating Characteristic (ROC) analysis, that these ENSO categories are in fact predictable with high accuracy, indicating that the conditioning is physically realistic rather than artificial (Fig. S7).

### Prediction method

Prediction skill is assessed using the temporal anomaly correlation coefficient (ACC) between the standardized ensemble-mean time series and observations. Anomalies are calculated by removing the trend and climatology from each model ensemble mean and the corresponding observations, following the 'Model-Climo Method' described in Meehl et al. (2022)[52]. ACC can also be interpreted as the predictable component embedded in the observations ($PC_o$).

In addition to ACC, two statistical prediction metrics are applied: the predictable component in models ($PC_m$) and the ratio of predictable components (RPC)[35,36]. $PC_m$ is defined as the square-rooted ratio of the ensemble mean variance (i.e., signal ($\sigma_{Sm}^2$)) to the mean variance across all ensemble members (i.e., total ($\sigma_{Tm}^2$) = signal ($\sigma_{Sm}^2$) + noise ($\sigma_{Nm}^2$)).

$$PC_m = \sqrt{\frac{\sigma_{Sm}^2}{\sigma_{Tm}^2}} = \sqrt{\frac{\sigma_{Sm}^2}{\sigma_{Sm}^2 + \sigma_{Nm}^2}} \quad (2)$$

RPC is calculated either as the ratio of $PC_o$ to $PC_m$, or equivalently as the ratio of the correlation between the ensemble mean and observations ($r_{mo}$) to the average correlation between the ensemble mean and individual members ($r_{mm}$).

$$RPC = \frac{PC_o}{PC_m} = \frac{r_{mo}}{r_{mm}} \quad (3)$$

The relative contribution (RC) quantifies the extent to which each individual year i contributes to the total correlation skill. It is calculated from the observations ($y_i$), ensemble-mean anomaly ($\bar{x}_i$), and their

standard deviations ($\sigma_y$, $\sigma_{\bar{x}}$)[19,53].

$$\text{RC}_i = \frac{\bar{x}_i y_i}{\sigma_{\bar{x}} \sigma_y} \tag{4}$$

The dependence of ACC skill on ensemble size is also theoretically estimated under a perfect-model assumption. The expected skill of an M-member ensemble mean ($C_M$) expressed[54]:

$$C_M = \frac{M^{1/2} C_1}{[1 + (M-1)C_1]^{1/2}} \tag{5}$$

where $M$ is the ensemble size, $C_1$ represents the skill of a single ensemble member.

## Bootstrap resampling

To estimate sampling uncertainty in MME, we employed a non-parametric bootstrap approach. For each statistic, 70 ensemble members were randomly resampled with replacement to generate 1000 pseudo-ensembles of size 70. The ensemble-mean value (e.g., NAO and ENSO index, ACC, SLP, or AAM composites) was computed for each pseudo-ensemble, yielding a bootstrap distribution of the statistic. Confidence intervals were defined using the 10–90th or 25–75th percentiles of the bootstrap distribution. The same bootstrap samples were used to assess grid-point statistical significance: stippling indicates locations where the bootstrap confidence interval does not include zero (95% confidence level in Fig. 2 and 90% level in Fig. 4). For clarity, only the bootstrap-mean field is shown in the figures. For the analysis of ensemble-size dependence in Fig. 5, we performed 10,000 bootstrap resamples without replacement to better sample the combinations of ensemble members for each ensemble size.

## Data availability

The original CMIP6 and DCPP database can be downloaded from the Earth System Grid Federation (ESGF) server (https://esgf-node.llnl.gov). The CESM2-SMYLE is obtained from the NCAR website (https://www.earthsystemgrid.org/dataset/ucar.cgd.cesm2.smyle.html). The HadSLP2 and HadISST1 are obtained from the Met Office Hadley Centre observations datasets (https://www.metoffice.gov.uk/hadobs/index.html). The ERA5 reanalysis data is freely available at Climate Data Store (https://cds.climate.copernicus.eu/).

## Code availability

The analysis code used in this study is publicly available at Figshare (https://doi.org/10.6084/m9.figshare.31429355).

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

## Acknowledgements

K.K. and M.-I.L. were funded by the Korea Meteorological Administration Research and Development Program "Operating and Developing Global Seasonal Forecast System" (KMA2018-00322) and "Grant RS-2025-02313090". A.A.S. and D.M.S. were supported by the Met Office Hadley Centre Climate Programme (HCCP) funded by the UK Department for Science, Innovation and Technology (DSIT) and the UK Public Weather Service.

## Author contributions

K.K. designed the research, performed the analysis, and wrote the original draft. M.-I.L., A.A.S., and D.M.S. contributed to the refinement of the methodology and provided comments to improve the manuscript. M.-I.L. supervised the project. All authors reviewed the final manuscript.

## Competing interests

The authors declare no competing interests.
