## [Transparent Peer Review file · Nature Communications]

ENSO phase transition enables prediction of winter North Atlantic Oscillation one year ahead

Corresponding Author: Professor Myong-In Lee

Version 0:

Reviewer comments:

Reviewer #1

(Remarks to the Author)

The paper by Kim et al. (2025) represents the first systematic validation, within an initialized multi-model and multi-year prediction system, of the enhancement effect of ENSO phase transitions on the prediction skill of the second-winter NAO. It also clarifies the relationship of this enhancement with the signal-to-noise paradox and ensemble size. Overall, the topic is novel, the approach is clear, and the findings are innovative. However, the manuscript also suffers from significant issues. Therefore, major revisions are required before the manuscript can be considered further.

Major Comments:

1. The central concept of the paper draws from Scaife et al. (2024), namely that ENSO can significantly influence the NAO with a one-year lag. I remain somewhat skeptical of this conclusion. The apparent impact of ENSO on the subsequent year's NAO inherently includes the concurrent influence of the tropical Pacific sea-air conditions during that second year. When these signals are blended, it is challenging to draw definitive conclusions. I believe a more detailed and in-depth analysis based on observational data is necessary. For instance, how does the NAO behave when the ENSO phase in the second year undergoes a transition, persists, or decays (i.e., becomes neutral in the second year)? This study finds significant NAO anomalies only during phase transitions, but not during persistence. Does this, from another perspective, suggest that the state of ENSO in the second year itself is more critical for the NAO? The crucial issue of how to more effectively disentangle the lagged and concurrent influences of ENSO requires careful consideration.

2. From the standpoint of practical forecasting, the relevance and implications of the paper's conclusions are potentially limited. In operational climate prediction, even if the first winter's ENSO state is known, accurately predicting whether the ENSO phase in the following year will transition or persist remains highly challenging, as it requires overcoming the spring predictability barrier. Given this inherent difficulty, what is the practical application value of the study's findings for NAO prediction?

Minor Comments

1. The definitions of "ENSO transition years" and "ENSO persistence years" need to be explicitly stated. Based on my understanding, an ENSO transition year should involve Year 1 being El Niño (or La Niña) and Year 2 being La Niña (or El Niño). An ENSO persistence year should involve both Year 1 and Year 2 being El Niño (or La Niña). The authors are requested to provide a table listing these specific events.

2. L43-44: 'While decadal prediction systems have shown significant skill in simulating multi-year mean NAO variability, ...'. Is this statement supported by references?

3. L45-47: 'A small number of systems have demonstrated skill in predicting the NAO one year ahead, highlighting the importance of large ensembles and external forcings beyond the initialized season'. This phrasing is somewhat difficult to follow. Please rephrase for clarity.

4. L69: Regarding the study period, Supplementary Table 1 indicates the period 1960-2018. Following the paper's logic, the period for the predicted Year-2 NAO should be 1962-2020. Why is it given as 2019 here? Furthermore, the authors label the

initialization month (November) as year -1, the next year as year 0, and the subsequent year as year 1, while also referring to the NAO in years 1 and 2. It is recommended to unify the terminology for the years to avoid potential confusion.

5. L82-83: 'This suggests that ENSO transitions after initialization are a major driver of improved YR2 NAO prediction'. I find this conclusion somewhat premature; at most, it can be stated that the two phenomena are concomitant.

6. L286-288: 'The MME ENSO index is predicted one year in advance under the assumption that the model world matches the observed ENSO phase. Only years with consistent ENSO characteristics in both worlds are used'. Does the term "consistent phase" here refer specifically to the ENSO state in Year 1?

Reviewer #2

(Remarks to the Author)

Based on previous work and insight on the delayed effect of ENSO forcing and slow poleward propagation of atmospheric angular momentum, this study evaluates the forecast skill of the North Atlantic Oscillation (NAO) in the second winter using a multi-model ensemble approach. It is found that the models can capture the atmospheric angular momentum propagation associated with the ENSO phase transition, and that the one-year lead NAO forecast skill is enhanced when there is an ENSO phase transition. This work demonstrates strong evidence of long-lead predictability of the NAO and provides support for the delayed effect of ENSO. It is a nice contribution to climate variability and prediction research. The analysis is carried out logically and quite thoroughly. I have a few comments and suggestions related to definition of target season and clarification. Some revisions are recommended before it is acceptable for publication.

Major comments:

1) The definition of winter season under study needs more justification or needs to be modified. In the present manuscript, winter NAO is defined as the average of January and February (JF) whereas ENSO with November-January (NDJ). For boreal winter, DJF is normally used. Although the variations are small, this gives the impression that the results are sensitive to such details. For seasonal forecast studies, especially multi-year seasonal forecasts like this one, no such sensitivity should be expected. I would suggest that for both winter ENSO and NAO, DJF is used. If the authors decide to keep using JF, probably "winter" could be replaced by "late winter" throughout the manuscript as well as in the title, but this would then make the topic close to a sub-seasonal problem.

2) I am confused with the bootstrap resampling approach. In the Methods section, it is mentioned in only one sentence (lines 323-324). My understanding is that it is used to estimate the statistical significance and confidence interval (as the error bars in Supplementary Fig. 1). However, in several figure captions, e.g., Figs. 2 and 4, "bootstrapping ensemble for 1000 times" is mentioned and there is no confidence intervals are given. What does this "bootstrapping ensemble" mean? More explanation on this should be given in the Methods section, or clarification should be made for those figure captions.

3) In the definition of ENSO transition, there is no amplitude requirement for NDJ (1) ENSO. This means that two cases with the same ENSO in NDJ (0) would possibly end up in different categories if they have a very small amplitude of ENSO in NDJ (1).

Minor comments:

1) Almost all the figure captions in the manuscript and in the Supplementary Information need edits and clarifications.

2) Lines 69-70: remove "tends to". The variability is severely underestimated.

3) Line 77: The correlation is not statistically significant after 2005. Any comments?

4) Line 80: How is the ENSO transition frequency defined?

5) Line 80: "peaks in the early 1960s". This is not shown in the figure.

6) Line 87: "Composites of NAO variability and skill...". What does this mean? Maybe using different wording.

7) Lines 118-122: The paragraph is unclear and does not read well. Please reword.

Version 1:

Reviewer comments:

Reviewer #1

(Remarks to the Author)

The authors have thoroughly addressed all of my concerns. I am now pleased to recommend that this work be accepted.

Reviewer #2

(Remarks to the Author)

I am satisfied with the response to my comments and the revision. I recommend acceptance of this revised manuscript for publication.

**Reviewer 1 Comment**

The paper by Kim et al. (2025) represents the first systematic validation, within an initialized
multi-model and multi-year prediction system, of the enhancement effect of ENSO phase
transitions on the prediction skill of the second-winter NAO. It also clarifies the relationship
of this enhancement with the signal-to-noise paradox and ensemble size. Overall, the topic is
novel, the approach is clear, and the findings are innovative. However, the manuscript also
suffers from significant issues. Therefore, major revisions are required before the manuscript
can be considered further.

We are grateful for the reviewers' careful comments. The following responses clarify the issues
raised about the figures and tables.

**Major Comments**

1. The central concept of the paper draws from Scaife et al. (2024), namely that ENSO can
significantly influence the NAO with a one-year lag. I remain somewhat skeptical of this
conclusion. The apparent impact of ENSO on the subsequent year's NAO inherently includes
the concurrent influence of the tropical Pacific sea-air conditions during that second year. When
these signals are blended, it is challenging to draw definitive conclusions. I believe a more
detailed and in-depth analysis based on observational data is necessary. For instance, how does
the NAO behave when the ENSO phase in the second year undergoes a transition, persists, or
decays (i.e., becomes neutral in the second year)? This study finds significant NAO anomalies
only during phase transitions, but not during persistence. Does this, from another perspective,
suggest that the state of ENSO in the second year itself is more critical for the NAO? The
crucial issue of how to more effectively disentangle the lagged and concurrent influences of
ENSO requires careful consideration.

Our aim is not to re-test Scaife et al. (2024)¹ but to show that NAO skill is enhanced during
transition years, consistent with their findings. The original paper by Scaife et al.¹ also
addressed the question of second winter influences from the Pacific, showing that composite
means of second winters contain relatively small anomalies from the concurrent ENSO events
(See Table 1 in Scaife et al. 2024¹). In there, the NAO anomalies in the following winter
increase significantly due to constructive interference in the transition years, such as +2.6 hPa
for the El Niño followed by La Niña, and -5.0 hPa for the La Niña followed by El Niño, which
magnitudes are larger than the concurrent anomaly differences between El Niño and La Niña
50 years.

Nevertheless, to further illustrate the key relationships, we re-examine the results by classifying
events according to ENSO phase evolutions and considering multiple ENSO evolution
pathways. Table R1 compares JF-mean NAO anomalies for cases in which El Niño or La Niña
conditions in NDJ(0) evolve into warm or cold SST states in NDJ(1), separating persistence
and transition cases. Statistically significant NAO anomalies emerge only in transition cases,
whereas persistence cases show no robust NAO response. This result is consistent with
previous studies and reinforces the central conclusion of this study that ENSO phase transitions,
rather than ENSO state alone, are critical for generating predictable NAO anomalies.

**Table R1:** Observed NAO anomaly changes between the first winter JF(0) and the second winter JF(1),
 composited by ENSO evolution. ENSO transition and persistence are defined by the sign of the ENSO index in
 NDJ(1) relative to NDJ(0), using 0 K as the threshold. Each number shows the number of years, the average
 Δ NAO (JF(1) - JF(0), hPa), and the associated p-value. Bold values indicate statistical significance at the 5% level.

ENSO state in NDJ(1)	La Niña (NDJ(0))	El Niño (NDJ(0))
positive (> 0 K)	8, -1.02, 0.04	7, 0.14, 0.74
negative (< 0 K)	12, 0.03, 0.82	13, 0.74, 0.04

Importantly, when NDJ(0)–ENSO is neutral and NDJ(1)–ENSO is El Niño or La Niña,
 meaning that only the concurrent Rossby-wave teleconnection to the NAO is active, this does
 not generate statistically significant NAO anomalies (Table S1). The result highlights that,
 beyond the concurrent ENSO teleconnection, the one-year-lagged dynamical pathway plays an
 important role in shaping NAO anomalies.

**Table S1:** Observed NAO anomaly changes between the first winter JF(0) and the second winter JF(1),
 composited by ENSO evolution. ENSO transition and persistence are defined by the sign of the ENSO index in
 NDJ(1) relative to NDJ(0), using 0 K as the threshold. Each number shows the number of years, the average
 Δ NAO (JF(1) - JF(0), hPa), and the associated p-value. Bold values indicate statistical significance at the 5% level.

ENSO state in NDJ(1)	La Niña (NDJ(0))	Neutral (NDJ(0))	El Niño (NDJ(0))
positive (> 0 K)	8, -1.02, 0.04	12, -0.01, 0.94	7, 0.14, 0.74
negative (< 0 K)	12, 0.03, 0.82	6, -0.77, 0.40	13, 0.74, 0.04

We added a paragraph in the main text with Table S1 (L118-122).

“This is consistent with composites of the observed NAO change one year later (Δ NAO), which
 show that when an ENSO event occurs in the preceding year and subsequently transitions, both
 El Niño and La Niña cases are associated with significant NAO anomalies. By contrast, no
 statistically significant NAO changes are observed when the preceding year is a neutral state
 ($|\text{ENSO}| < 0.5$ K) or when ENSO persists into the following year (Table S1).”

2. From the standpoint of practical forecasting, the relevance and implications of the paper's
 conclusions are potentially limited. In operational climate prediction, even if the first winter's
 ENSO state is known, accurately predicting whether the ENSO phase in the following year will
 transition or persist remains highly challenging, as it requires overcoming the spring
 predictability barrier. Given this inherent difficulty, what is the practical application value of
 the study's findings for NAO prediction?

As pointed out by the reviewer, the spring predictability barrier of ENSO remains a major
 challenge that can limit prediction skill. However, as mentioned in L311-319 in the main text,
 most models actually show a reasonable level of skill for predicting ENSO one year ahead
 ($\text{ACC} \sim 0.5$). This has been demonstrated in Dunstone et al. (2020)² and Knight et al. (2014)³,
 and the skill is particularly high for strong events (e.g., Luo et al., 2017⁴). Therefore, the idea
 that such predictions could be useful in real forecasting applications remains plausible,
 although we emphasize that this paper focuses on understanding the sources of interannual
 prediction skill rather than operational real-time forecasting.

To clarify this further, we examined the actual ENSO forecast skill in greater detail using the
 decadal prediction systems and the MME used in this study. We found statistically significant
 ENSO prediction skill one year, extending beyond the spring predictability barrier, with an
 overall ACC of approximately 0.5–0.6 over the full period (Fig. S6a; $p < 0.01$). Moreover, the
 model ensemble world shows clear and statistically significant skill in predicting the observed
 transition (21 years) and persistence (19 years) characteristics (Figs. S6b–c).

**Fig. S6: Prediction skill of the ENSO index in all, ENSO transition, and persistence years.**

(a) ACC of the 3-month-mean Niño3.4 index as a function of lead time (up to 15 months) for the MME (black)
 and individual decadal prediction systems (colored lines) during 1962-2019. The thick black dashed curve
 denotes the persistence forecast based on the November Niño3.4 index, and the horizontal dashed line indicates
 the 5% significance level. Shading around the MME shows the ensemble spread between the 10th and 90th
 percentiles estimated from 1,000 bootstrap resamples. (b) Same as (a), but composited for ENSO transition years.
 (c) Same as (a), but composited for ENSO persistence years.

Because this study emphasizes that whether ENSO undergoes a phase transition or persists into
 the following year is critical for NAO predictability, evaluating ENSO forecast skill solely

using ACC may be insufficient. We therefore further assess the probabilistic prediction skill of
 NDJ(1) ENSO using tercile-based categories (upper, middle, and lower terciles). In the
 subsequent analysis, we classified ENSO transition and persistence using cases in which ENSO
 is strongly developed at initialization (threshold = ± 0.5 K), and evaluate the probabilistic
 prediction skill for the second winter by computing area under the curve (AUC) values for El
 Niño and La Niña based on tercile thresholds, conditioned on years in which the first winter
 exhibits observed El Niño or La Niña conditions (Fig. S7). This analysis directly assesses the
 system’s ability to predict whether ENSO undergoes a transition or persists into the second
 winter. The AUC values for both transition and persistence cases exceed 0.5, indicating high
 hit rates, with particularly strong skill for transition events, especially for El Niño-to-La Niña
 (AUC = 0.86) and La Niña-to-El Niño (AUC = 0.95) transitions.

 **Fig. S7:** Receiver Operating Characteristic (ROC) curves for the NDJ(1) ENSO index predicted by the MME,
 evaluated for (a) ENSO transition years and (b) ENSO persistence years. The dashed diagonal line represents no-
 skill prediction. Area under the curve (AUC) values are shown in the legends, quantifying the forecast skill of the
 MME.

 The ENSO transition characteristics can therefore serve as a practical predictor for the one-
 131 year-ahead NAO forecasts. Moreover, because persistence is also predictable, forecasters can
 explicitly account for the greater uncertainty associated with persistence years and adopt a more
 conservative interpretation of NAO forecasts. In this sense, providing probabilistic information
 on ENSO phase evolution offers a practical framework for identifying when NAO predictions
 are likely to be reliable and when they should be treated with caution.

Based on the additional results in the above, we improved the main text (L26-27; L250-257;
 and L323-328), with newly added Figs. S6 and S7.

“.... , given the skillful prediction of ENSO phase transitions at one-year lead times by multi-
 model ensembles” (L26-27).

“The MME results indicate that NAO prediction skill is enhanced during ENSO phase
 transitions. For this result to be practically meaningful, it is essential to assess how well climate

models can predict ENSO phase evolution (see Methods). The MME exhibits statistically
 significant long-range ENSO prediction skill through the LY1 winter despite the spring
 predictability barrier (Fig. S6). We further examine one-year-ahead tercile prediction skill
 conditioned on the initial ENSO state (Fig. S7). We demonstrated that these categories are
 themselves predicted with high accuracy, confirming that the conditioning reflects realistic
 forecast skill rather than an artificial perfect-model assumption.” (L250-257).

“For the NAO-skill composites, we conditioned on years in which the model-predicted NDJ(1)
 ENSO category (transition or persistence) agrees with matched observations (Fig. S10). This
 so-called ‘perfect-model’ filtering was applied solely as a diagnostic tool to isolate the
 underlying physical mechanisms. We further demonstrate, using Relative Operating
 Characteristic (ROC) analysis, that these ENSO categories are in fact predictable with high
 accuracy, indicating that the conditioning is physically realistic rather than artificial (Fig. S7).”
 (L323-328).

**Minor Comments**

1. The definitions of "ENSO transition years" and "ENSO persistence years" need to be
 explicitly stated. Based on my understanding, an ENSO transition year should involve Year 1
 being El Niño (or La Niña) and Year 2 being La Niña (or El Niño). An ENSO persistence year
 should involve both Year 1 and Year 2 being El Niño (or La Niña). The authors are requested
 to provide a table listing these specific events.

Following the comment, we add “ENSO phase change definitions” part in Methods section
 (L309). We also provide the information of years for those specific events in Table S4.

**Table S4:** Classification of ENSO phase evolution and composite years used in this study. ENSO phase changes
 are defined based on transitions between NDJ(0) and NDJ(1). Numbers in parentheses indicate the total number
 of years in each category, and listed years denote the corresponding calendar years.

ENSO phase changes from NDJ(0) to NDJ(1)		
ENSO regime	ENSO evolution	Years
Transition (21)	El Niño → La Niña (9)	1965, 1971, 1974, 1989, 1996, 1999, 2006, 2008, 2011
	El Niño → weak La Niña (4)	1967, 1981, 1984, 2017
	La Niña → El Niño (8)	1966, 1969, 1973, 1977, 1987, 2007, 2010, 2019
	La Niña → weak El Niño (0)	N/A
Persistence (19)	El Niño → El Niño (4)	1970, 1978, 1988, 2016
	El Niño → weak El Niño (3)	1979, 1993, 2004

La Niña → La Niña (8)	1972, 1975, 1976, 1986, 2000, 2001, 2009, 2012
La Niña → weak La Niña (4)	1990, 1997, 2002, 2013

2. L43-44: 'While decadal prediction systems have shown significant skill in simulating multi-
169 year mean NAO variability, ...'. Is this statement supported by references?

We added references to Smith et al. (2020)⁵, Athanasiadis et al. (2020)⁶ papers (L45).

3. L45-47: 'A small number of systems have demonstrated skill in predicting the NAO one year
ahead, highlighting the importance of large ensembles and external forcings beyond the
initialized season'. This phrasing is somewhat difficult to follow. Please rephrase for clarity.

We revise this sentence.

" A few systems have demonstrated significant skill in predicting the NAO one year ahead and
they have shown that importance of the ensemble size (40 members) and incorporating external
dynamical factors such as sea surface temperature (SST) oceanic variability, ENSO, North
Atlantic Tripolar SST (NATS), sea ice, and a strong polar vortex (SPV) through remote
teleconnections can enhance NAO prediction skill." (L46-51).

4. L69: Regarding the study period, Supplementary Table 1 indicates the period 1960-2018.
Following the paper's logic, the period for the predicted Year-2 NAO should be 1962-2020.
Why is it given as 2019 here? Furthermore, the authors label the initialization month
(November) as year -1, the next year as year 0, and the subsequent year as year 1, while also
referring to the NAO in years 1 and 2. It is recommended to unify the terminology for the years
to avoid potential confusion.

We appreciate the comment. In the previous version of the manuscript, there was a discrepancy
between the stated hindcast initialization period and the actual analysis period. The DCP
hindcasts are indeed generated by initializing each November from 1960 to 2018, which in
principle allows NAO predictions to be evaluated for the period 1962–2020. However, because
the Hadley Centre Sea Level Pressure version 2 (HadSLP2) dataset used to compute the NAO
index is only available through December 2019, we excluded the year 2020 from the analysis.

Accordingly, we corrected the information on the hindcast period in Table S2 from “1960–
2018” to “1960–2017” to ensure consistency with the analysis period. We note that the 2019–
2020 period corresponds to an El Niño persistence year, which does not contribute to enhanced
NAO prediction skill in the same way as transition years. Nevertheless, even when the 2020
NAO index from ERA5 is included in the evaluation, the overall results remain unchanged.
Over the extended period 1962–2020, the overall NAO prediction skill is 0.12 ($p > 0.05$). When
stratified by ENSO behavior, the skill during persistence years (17 years) is -0.13 ($p > 0.05$),
whereas the skill during transition years (17 years) remains high at 0.60 ($p < 0.05$).

In S2D predictions initialized in November, the first and second winters were referred to as
 YR1 (lead < 3 months) and YR2 (lead < 15 months), respectively, in the original manuscript.
 To avoid inconsistent terminology, we standardize definitions such that the first winter
 corresponds to lead year 0 (LY0) and JF(0), and the second winter to lead year 1 (LY1) and
 JF(1).

“...., to investigate the predictability of the NAO during the second winter after initialization
 (i.e., one-year lead; hereafter LY1).” (L55-56)

“We defined the first boreal winter after initialization as lead year 0 (LY0) and the second
 boreal winter as LY1. For example, for a forecast initialized in November 1980, LY0
 corresponds to the JF(0) winter of 1981 (2-3 month lead), and LY1 corresponds to the JF(1)
 winter of 1982 (14-15 month lead). This study focused on evaluating NAO prediction skill
 during LY1.” (L270–274).

5. L82-83: 'This suggests that ENSO transitions after initialization are a major driver of
 improved YR2 NAO prediction'. I find this conclusion somewhat premature; at most, it can be
 stated that the two phenomena are concomitant.

As discussed in detail in **Major Comment 1**, the delayed ENSO–NAO dynamics identified
 here are physically robust. When the ENSO phase was neutral on first winter (NDJ(0)), no
 significant change in the NAO anomaly occurred for the concomitant ENSO phase (Table R2).
 This insight is the importance of the effect of atmospheric angular momentum propagating one
 222 year prior. Our findings are consistent with ENSO transitions being a major driver of year ahead
 prediction skill for the NAO.

**Table R2:** Same as in Table S1 but now the NDJ(1) ENSO states are separated into three categories of El Niño
 (>0.5 K), neutral, and La Niña (<-0.5K) in long-term period (1872-2019). Bold text means 5% significance level
 is indicated.

ENSO state in NDJ(1)	La Niña (NDJ(0))	Neutral (NDJ(0))	El Niño (NDJ(0))
positive (> 0.5 K)	20, -0.72, 0.02	14, 0.28, 0.30	11, 0.16, 0.60
neutral	10, -0.32, 0.39	23, 0.00, 0.99	19, 0.57, 0.07
negative (< -0.5 K)	22, 0.05, 0.87	15, -0.64, 0.12	14, 0.67, 0.04

6. L286-288: 'The MME ENSO index is predicted one year in advance under the assumption
 that the model world matches the observed ENSO phase. Only years with consistent ENSO
 characteristics in both worlds are used'. Does the term "consistent phase" here refer specifically
 to the ENSO state in Year 1?

No. Here, “consistent ENSO characteristics” refers not to the ENSO state in the first winter
 (LY0), but to the ENSO transition or persistence behavior in the second winter (LY1). We
 include only those years in which the model reproduces the observed ENSO evolution
 (transition or persistence) in LY1. This conditioning is used solely as a diagnostic to isolate the
 physical mechanism and is justified by the high hit rates and significant prediction skill of
 ENSO phase evolution in the models (Fig. S1b, Fig. S6, and Fig. S7). We clarified this in the
 revised manuscript.

“To evaluate NAO prediction skill one year ahead (LY1), composites were constructed based
on ENSO phase evolution. The MME ENSO index was predicted one year in advance, and
only those years in which the ENSO evolution category in LY1 matched between observations
and the MME were retained. This conditioning ensures that the composite analysis isolates
NAO predictability associated with correctly simulated ENSO phase evolution. Although this
approach does not explicitly address the spring predictability barrier, the agreement in ENSO
category between observations and the MME remains significant prediction skill through LY1,
supporting the validity of the composite framework (17 out of 21 transition years and 16 out
of 19 persistence years in observations; Fig. S1b, S6).” (L314-322).

**Reviewer 2 Comment**

Based on previous work and insight on the delayed effect of ENSO forcing and slow poleward
propagation of atmospheric angular momentum, this study evaluates the forecast skill of the
North Atlantic Oscillation (NAO) in the second winter using a multi-model ensemble approach.
It is found that the models can capture the atmospheric angular momentum propagation
associated with the ENSO phase transition, and that the one-year lead NAO forecast skill is
enhanced when there is an ENSO phase transition. This work demonstrates strong evidence of
long-lead predictability of the NAO and provides support for the delayed effect of ENSO. It is
a nice contribution to climate variability and prediction research. The analysis is carried out
logically and quite thoroughly. I have a few comments and suggestions related to definition of
target season and clarification. Some revisions are recommended before it is acceptable for
publication.

Thank you for the great comments. We address points of confusion regarding the figures and
revised sentences below.

**Major comments:**

1. The definition of winter season under study needs more justification or needs to be modified.
In the present manuscript, winter NAO is defined as the average of January and February (JF)
whereas ENSO with November-January (NDJ). For boreal winter, DJF is normally used.
Although the variations are small, this gives the impression that the results are sensitive to such
details. For seasonal forecast studies, especially multi-year seasonal forecasts like this one, no
such sensitivity should be expected. I would suggest that for both winter ENSO and NAO, DJF
is used. If the authors decide to keep using JF, probably “winter” could be replaced by “late
winter” throughout the manuscript as well as in the title, but this would then make the topic
close to a sub-seasonal problem.

We appreciate the reviewer’s thoughtful comments regarding the definition of the winter season.
We understand the concern that defining ENSO as NDJ and NAO as JF might imply a
sensitivity of our results to specific month selections. We note, however, that our choice was
driven by the distinct dynamical characteristics of the winter season and the aim to capture the
maximum potential predictability. In the revised manuscript, we added Fig. S2. In Fig. S2 when
predicting the DJF-mean NAO, although the absolute skill score is slightly lower than for the
JF-mean (likely due to the inclusion of the less active December month), the enhancement of
prediction skill one year ahead during ENSO transition years remains statistically significant
and robust.

**Fig. S2:** Same as in Fig. 1 except using the DJF-mean NAO index prediction.

The slight differences in skill obtained when using DJF means arise because this averaging
 does not optimally represent the seasons of maximum variability for ENSO and the NAO. As
 shown in Fig. S8 (below), the seasonal cycle of variability indicates that the ENSO signal peaks
 in NDJ, while the NAO variability reaches its maximum in JF. The DJF-mean and NDJ-mean
 ENSO indices have a correlation of 0.99, and the composite years associated with transition
 and persistence characteristics remain exactly the same for both definitions. Previous studies,
 such as Oji et al. (2004)⁷ and Lee et al. (2025)⁸ and Kryjov (2003)⁹, have highlighted that the
 Arctic Oscillation (AO/NAO) exhibits its most active and annular-mode-like features during
 January and February. Therefore, targeting JF ensures we are analyzing the most robust
 dynamical component of the winter circulation.

**Fig. S8:** Seasonal evolution of NAO and ENSO variability during 1962–2019. (a) Monthly standard deviation of
 the NAO index. Red circles denote the NAO index following Jones et al. (1997)¹⁰, while the blue line shows the
 EOF-based NAO index following Hurrell and Deser (2010)¹¹. (b) Monthly standard deviation of the ENSO index.
 Gray shaded boxes indicate the target seasons during which variability is maximized and which are used for the
 main analysis: January–February (JF) for the NAO and November–December–January (NDJ) for ENSO.

Furthermore, Ayarzagüena et al. (2018)¹² and Li and Lau (2012)¹³ demonstrated that ENSO
 teleconnections to the North Atlantic are not uniform throughout the winter; the response in
 early winter (ND) often differs from, or even opposes, the response in late winter (JF).
 Averaging these distinct periods into a single DJF-mean can obscure the specific delayed
 dynamical signal (e.g., via the stratosphere or AAM) we aim to predict. This intra-seasonal
 early-winter noise can weaken the clear linear relationships seen in Fig. 3, as shown in Fig. R1,
 which presents the observational scatter plots using DJF-mean ENSO and DJF-mean NAO
 indices.

**Fig. R1.** Same as in Fig. 3 except for using the DJF-mean ENSO and the DJF-mean NAO indices.

In the revised manuscript, we added a brief statement in the Fig. 1 in **Results** section regarding
 the DJF-mean L100-103.

“Using seasonal DJF-mean NAO indices yields qualitatively identical results, although with
 lower prediction skill than the JF-mean definition (Fig. S2), confirming that our conclusions
 are not sensitive to the specific winter season definition (see Methods).” (L100-103)

Regarding terminology, since the JF period accounts for the dominant portion of wintertime
 variance and the associated dynamical variability, we believe maintaining the term "Winter" is
 appropriate. Using "Late Winter" might inadvertently categorize this study as a sub-seasonal
 prediction problem. However, this study focuses on interannual predictability (one year lead),
 where the precise monthly definition is less about sub-seasonal timing and more about
 capturing the seasonal active mean state.

2. I am confused with the bootstrap resampling approach. In the Methods section, it is
 mentioned in only one sentence (lines 323-324). My understanding is that it is used to estimate
 the statistical significance and confidence interval (as the error bars in Supplementary Fig. 1).
 However, in several figure captions, e.g., Figs. 2 and 4, “bootstrapping ensemble for 1000 times”
 is mentioned and there is no confidence intervals are given. What does this “bootstrapping
 ensemble” mean? More explanation on this should be given in the Methods section, or
 clarification should be made for those figure captions.

In this study the bootstrap resampling is used to quantify sampling uncertainty of the

multi-model ensemble and to estimate confidence intervals and statistical significance (Kang
and Lee, 2017¹⁴). Specifically, for each statistic we construct 1,000 “bootstrap ensembles” by
randomly drawing 70 members with replacement from the original 70-member ensemble,
computing the ensemble-mean value (e.g. NAO and ENSO index, ACC, SLP or AAM
composite) for each resampled ensemble, and forming the empirical distribution of that statistic
from the 1,000 realizations.

From this bootstrap distribution we then: take the 10th–90th or 25th–75th percentiles to define
the shaded confidence intervals in Fig. 1, Fig. 5 and Supplementary Fig. S1, and determine
grid-point significance in the maps in Figs. 2 and 4: in those figures, stippling marks locations
where the bootstrap confidence interval does not include zero (95% level in Fig. 2 and 90%
level in Fig. 4).

Thus, the phrase “bootstrapping ensemble for 1,000 times” in the captions of Figs. 2 and 4
refers to this resampling of the 70-member ensemble that is used to assess the robustness of the
composite patterns and skill maps. For clarity, we have revised the captions to explicitly state
that bootstrapping is used to estimate significance, and that only the bootstrap-mean field (not
the full spread) is plotted in these panels to avoid clutter.

We have also expanded the Methods section (new “Bootstrap resampling” paragraph) to
describe the procedure and its different uses for scalar metrics (time series and skill measures)
and for spatial fields (composite maps), so that the reader can clearly see how the bootstrap is
applied throughout the paper.

**“Bootstrap resampling**

To estimate sampling uncertainty in MME, we employed a non-parametric bootstrap approach.
For each statistic, 70 ensemble members were randomly resampled with replacement to
generate 1,000 pseudo-ensembles of size 70. The ensemble-mean value (e.g. NAO and ENSO
index, ACC, SLP or AAM composites) was computed for each pseudo-ensemble, yielding a
bootstrap distribution of the statistic. Confidence intervals were defined using the 10–90th or
25–75th percentiles of the bootstrap distribution. The same bootstrap samples were used to
assess grid-point statistical significance: stippling indicates locations where the bootstrap
confidence interval does not include zero (95% confidence level in Fig. 2 and 90% level in Fig.
4). For clarity, only the bootstrap-mean field is shown in the figures. For the analysis of
ensemble size dependence in Fig. 5, we performed 10,000 bootstrap resamples without
replacement to better sample the combinations of ensemble members for each ensemble size.”
(L354-365)

**3. In the definition of ENSO transition, there is no amplitude requirement for NDJ (1) ENSO.**
**This means that two cases with the same ENSO in NDJ (0) would possibly end up in different**
**categories if they have a very small amplitude of ENSO in NDJ (1).**

This comment overlaps with Reviewer 1’s first major comment. In this study, ENSO transition
and persistence are classified based on the sign of SST anomalies in NDJ(1). As correctly noted
by the reviewer, this definition does not represent a strict phase transition in the dynamical
sense. This choice was made to increase sample size and allow robust statistical evaluation.

Importantly, we verified that the main conclusions are insensitive to this definition. When
 NDJ(1) ENSO states are instead categorized explicitly into El Niño, neutral, and La Niña
 conditions, the results remain fully consistent with those presented in the manuscript.

Table R1 reproduces the analysis in the main text by comparing JF-mean NAO anomalies for
 cases in which El Niño or La Niña conditions in NDJ(0) evolve into warm or cold SST states
 in NDJ(1), separating transition and persistence cases. Statistically significant NAO anomalies
 emerge only in transition cases, whereas persistence cases show no robust NAO response. This
 result is consistent with the main conclusion of this work.

Moreover, we repeated the analysis using a ± 0.5 K threshold to identify strong transition and
 persistence cases in NDJ(1) ENSO states (Table R3). Under this stricter definition, transitions
 from La Niña to El Niño produce a significant and strongly negative Δ NAO anomaly. In
 contrast, El Niño to La Niña transitions yield a weaker and statistically insignificant response
 within the original analysis period. This asymmetry is likely influenced by sampling limitations,
 as El Niño to La Niña transitions become statistically significant when the evaluation period is
 extended to 1872–2019 (Table R4).

**Table R1:** Observed NAO anomaly changes between the first winter JF(0) and the second winter JF(1),
 composited by ENSO evolution. ENSO transition and persistence are defined by the sign of the ENSO index in
 NDJ(1) relative to NDJ(0), using 0 K as the threshold. Each number shows the number of years, the average
 Δ NAO (JF(1) - JF(0), hPa), and the associated p-value. Bold values indicate statistical significance at the 5% level.

ENSO state in NDJ(1)	La Niña (NDJ(0))	El Niño (NDJ(0))
positive (> 0 K)	8, -1.02, 0.04	7, 0.14, 0.74
negative (< 0 K)	12, 0.03, 0.82	13, 0.74, 0.04

**Table R3.** Same as in **Table R1** but now the NDJ(1) ENSO states are separated into three categories of El Niño
 (>0.5 K), neutral, and La Niña (<-0.5 K). Bold text means 5% significance level is indicated.

ENSO state in NDJ(1)	La Niña (NDJ(0))	El Niño (NDJ(0))
positive (> 0.5 K)	8, -1.02, 0.04	4, 0.53, 0.51
neutral	4, 0.12, 0.76	7, 0.50, 0.33
negative (< -0.5 K)	8, -0.02, 0.95	9, 0.55, 0.19

**Table R4.** Same as in **Table R3** but for the long-term (1872-2019) period.

ENSO state in NDJ(1)	La Niña (NDJ(0))	El Niño (NDJ(0))
positive (> 0.5 K)	20, -0.72, 0.02	11, 0.16, 0.60
neutral	10, -0.32, 0.39	19, 0.57, 0.07
negative (< -0.5 K)	22, 0.05, 0.87	14, 0.67, 0.04

Further discussion of this point can be found in our response to Reviewer 1’s Major Comment
 1. We added a paragraph in the main text with Table S1 (L118-122).

“This is consistent with composites of the observed NAO change anomaly one year later
 (Δ NAO), which show that when an ENSO event occurs in the preceding year and subsequently
 transitions, both El Niño and La Niña cases are associated with significant NAO anomalies. By
 contrast, no statistically significant NAO changes are observed when the preceding year is a

neutral state ($|\text{ENSO}| < 0.5 \text{ K}$) or when ENSO persists in the following year (Table S1).”

**Table S1:** Observed NAO anomaly changes between the first winter JF(0) and the second winter JF(1),
composited by ENSO evolution. ENSO transition and persistence are defined by the sign of the ENSO index in
NDJ(1) relative to NDJ(0), using 0 K as the threshold. Each number shows the number of years, the average
ΔNAO (JF(1) - JF(0), hPa), and the associated p-value. Bold values indicate statistical significance at the 5% level.

ENSO state in NDJ(1)	La Niña (NDJ(0))	Neutral (NDJ(0))	El Niño (NDJ(0))
positive ($> 0 \text{ K}$)	8, -1.02, 0.04	12, -0.01, 0.94	7, 0.14, 0.74
negative ($< 0 \text{ K}$)	12, 0.03, 0.82	6, -0.77, 0.40	13, 0.74, 0.04

**Minor Comments**

1) Almost all the figure captions in the manuscript and in the Supplementary Information need
edits and clarifications.

We have carefully and thoroughly revised all the figure captions in the manuscript and in the
Supplementary Information (Since we’ve revised all the captions, we didn't mark the text in
red.).

2) Lines 69-70: remove “tends to”. The variability is severely underestimated.

We revise the sentence.

“The ensemble spread of NAO anomalies is large, and the MME severely underestimates the
observed amplitude of the index (Fig. 1a)” (L73-74).

3) Line 77: The correlation is not statistically significant after 2005. Any comments?

This ACC skill reduction lies well within sampling uncertainty. Given the limited number of
cases in the post-2005 period, particularly when further subdividing into ENSO transition and
persistence events, the apparent decrease in skill is not statistically robust. We therefore refrain
from over-interpreting temporal variations in correlation over such a short period. Our main
conclusions are based on the full analysis period, where the signal remains statistically
significant and physically consistent with the proposed mechanism in composited ENSO
transition period.

4) Line 80: How is the ENSO transition frequency defined?

Following the comment, we add “ENSO phase change definitions” part in Methods section
(L306). We also provide the information of years for those specific events in Table S4. The
ENSO transition frequency is defined as a binary time series, with a value of 1 assigned to
440 years in which an ENSO transition occurs (indicated by crosses along the bottom of Fig. 1d)
and 0 otherwise. The dashed line in Fig. 1c represents the 23-year sliding mean of this binary

ENSO transition time series. We also improved the caption of Fig. 1.

**Table S4:** Classification of ENSO phase evolution and composite years used in this study. ENSO phase changes
 are defined based on transitions between NDJ(0) and NDJ(1). Numbers in parentheses indicate the total number
 of years in each category, and listed years denote the corresponding calendar years.

ENSO phase changes from NDJ(0) to NDJ(1)		
ENSO regime	ENSO evolution	Years
Transition (21)	El Niño → La Niña (9)	1965, 1971, 1974, 1989, 1996, 1999, 2006, 2008, 2011
	El Niño → weak La Niña (4)	1967, 1981, 1984, 2017
	La Niña → El Niño (8)	1966, 1969, 1973, 1977, 1987, 2007, 2010, 2019
	La Niña → weak El Niño (0)	N/A
Persistence (19)	El Niño → El Niño (4)	1970, 1978, 1988, 2016
	El Niño → weak El Niño (3)	1979, 1993, 2004
	La Niña → La Niña (8)	1972, 1975, 1976, 1986, 2000, 2001, 2009, 2012
	La Niña → weak La Niña (4)	1990, 1997, 2002, 2013

5) Line 80: “peaks in the early 1960s”. This is not shown in the figure.

The reference to “the 1960s” was a typo. We revised the sentence.

“The 23-year sliding-mean ENSO transition frequency (dashed line in Fig. 1c) exhibits peaks
 in the early **1970s** and after the mid-1990s, closely aligned with the 23-year running-mean
 NAO skill ($r = 0.94$, $p < 0.01$)” (L84-86).

6) Line 87: “Composites of NAO variability and skill...”. What does this mean? Maybe using
 different wording.

We clarified it in the sentence.

“Separating ENSO transition and persistence years reveals a clear contrast in NAO prediction
 skill (ACC in Fig. 1e, f). During transition years, observed NAO variability....” (L91-92).

...

7) Lines 118-122: The paragraph is unclear and does not read well. Please reword.

We revised the paragraph.

“These results demonstrate that the delayed ENSO–NAO teleconnection mechanism during
transition years is robust in both observations and models. In contrast, persistence years fail to
produce comparable NAO responses one year ahead and are poorly predicted, consistent with
weaker predictable signals and destructive interference between lagged and concurrent
teleconnections during ENSO persistence years” (L129-134).

**References**

- 1. Scaife AA, *et al.* ENSO affects the North Atlantic Oscillation 1 year later. *Science* **386**, 82–86
(2024).
- 2. Dunstone N, *et al.* Skilful interannual climate prediction from two large initialised model
ensembles. *Environmental Research Letters* **15**, (2020).
- 3. Knight JR, *et al.* Predictions of climate several years ahead using an improved decadal
prediction system. *Journal of Climate* **27**, 7550–7567 (2014).
- 4. Luo JJ, Liu G, Hendon H, Alves O, Yamagata T. Inter-basin sources for two-year predictability
of the multi-year La Nina event in 2010-2012. *Sci Rep* **7**, 2276 (2017).
- 5. Smith DM, *et al.* North Atlantic climate far more predictable than models imply. *Nature* **583**,
796–800 (2020).
- 6. Athanasiadis PJ, Yeager S, Kwon Y-O, Bellucci A, Smith DW, Tibaldi S. Decadal predictability
of North Atlantic blocking and the NAO. *npj Climate and Atmospheric Science* **3**, (2020).
- 7. Ogi M, Yamazaki K, Tachibana Y. The summertime annular mode in the Northern Hemisphere
and its linkage to the winter mode. *Journal of Geophysical Research: Atmospheres* **109**,
(2004).
- 8. Lee S, Lee M-I, Lee S, Jin F-F, Choi N, Tak S. Distinct Features of the Summer Arctic Oscillation.
*Journal of Climate* **38**, 5983–5996 (2025).
- 9. Kryjov VN. Searching for circulation patterns affecting North Europe annual temperature.
*Atmospheric Science Letters* **5**, 23–34 (2004).
- 10. Jones PD, Jonsson T, Wheeler D. Extension to the North Atlantic oscillation using early
instrumental pressure observations from Gibraltar and south-west Iceland. *International*
*Journal of Climatology* **17**, 1433–1450 (1997).

- 11. Hurrell JW, Deser C. North Atlantic climate variability: The role of the North Atlantic
Oscillation. *Journal of Marine Systems* **79**, 231–244 (2010).
- 12. Ayarzagüena B, Ineson S, Dunstone NJ, Baldwin MP, Scaife AA. Intraseasonal Effects of El
Niño–Southern Oscillation on North Atlantic Climate. *Journal of Climate* **31**, 8861–8873
(2018).
- 13. Li Y, Lau N-C. Impact of ENSO on the atmospheric variability over the North Atlantic in late
winter—Role of transient eddies. *Journal of Climate* **25**, 320–342 (2012).
- 14. Kang D, Lee M-I. Increase in the potential predictability of the Arctic Oscillation via intensified
teleconnection with ENSO after the mid-1990s. *Climate Dynamics* **49**, 2147–2160 (2017).